# The Role of BAR Proteins and the Glycocalyx in Brain Endothelium Transcytosis

**DOI:** 10.3390/cells9122685

**Published:** 2020-12-14

**Authors:** Diana M. Leite, Diana Matias, Giuseppe Battaglia

**Affiliations:** 1Department of Chemistry, University College London, London WC1H 0AJ, UK; d.leite@ucl.ac.uk (D.M.L.); d.matias@ucl.ac.uk (D.M.); 2Institute of the Physics and Living Systems, University College London, London WC1H 0AJ, UK; 3Samantha Dickson Brain Cancer Unit, Cancer Institute, University College London, London WC1E 06DD, UK; 4Cancer Research UK, City of London Centre, London WC1E 06DD, UK; 5Institute for Bioengineering of Catalonia (IBEC), The Barcelona Institute for Science and Technology (BIST), 08028 Barcelona, Spain; 6Catalan Institute for Research and Advanced Studies, 08010 Barcelona, Spain

**Keywords:** blood–brain barrier, endothelium, transcytosis, tubulation, BAR proteins, glycocalyx

## Abstract

Within the brain, endothelial cells lining the blood vessels meticulously coordinate the transport of nutrients, energy metabolites and other macromolecules essential in maintaining an appropriate activity of the brain. While small molecules are pumped across specialised molecular transporters, large macromolecular cargos are shuttled from one side to the other through membrane-bound carriers formed by endocytosis on one side, trafficked to the other side and released by exocytosis. Such a process is collectively known as transcytosis. The brain endothelium is recognised to possess an intricate vesicular endosomal network that mediates the transcellular transport of cargos from blood-to-brain and brain-to-blood. However, mounting evidence suggests that brain endothelial cells (BECs) employ a more direct route via tubular carriers for a fast and efficient transport from the blood to the brain. Here, we compile the mechanism of transcytosis in BECs, in which we highlight intracellular trafficking mediated by tubulation, and emphasise the possible role in transcytosis of the Bin/Amphiphysin/Rvs (BAR) proteins and glycocalyx (GC)—a layer of sugars covering BECs, in transcytosis. Both BAR proteins and the GC are intrinsically associated with cell membranes and involved in the modulation and shaping of these membranes. Hence, we aim to summarise the machinery involved in transcytosis in BECs and highlight an uncovered role of BAR proteins and the GC at the brain endothelium.

## 1. Introduction

Blood vessels in the brain are organised with a surprising precision, supporting the major neural circuits responsible for sensation, cognition, memory and motion [1]. Proper structural and functional brain connectivity requires a precise regulation of the cerebral blood flow, oxygen delivery and energy metabolite supply [1,2]. Within brain blood vessels, the endothelium tightly regulates the transport of molecules from blood-to-brain and brain-to-blood. Nevertheless, to facilitate the entry of nutrients, energy metabolites and other essential macromolecules from blood-to-brain and rid the brain of waste products, brain endothelial cells (BECs) express multiple substrate-specific transport systems [3,4,5]. Thus, BECs facilitate transport of essential molecules through vesicular/tubular membrane carriers—transcytosis. Steps along transcytosis include endocytosis, intracellular trafficking and exocytosis. In the brain endothelium, the machinery involved in transcytosis remains to be fully comprehended. By analogy to the epithelium, it is generally established that the trafficking across the brain endothelium occurs via vesicular carriers. Nevertheless, recent findings suggest that BECs employ sorting tubules as a fast mechanism of transcytosis, in which the cargo is efficiently transported across via tubular carriers, avoiding endosomes and lysosomal degradation [6,7]. Interestingly, these results corroborate evidence from the 1970s and 1990s, in which it was suggested that intracellular trafficking occurs through a network of tubules at the brain endothelium [8,9,10,11,12].

During trafficking, cell membranes become highly curved to allow the formation of the vesicular and tubular carriers from flat membranes. One of the best-known regulators of membrane curvature is a family of proteins that comprises the crescent-shaped Bin/Amphiphysin/Rvs (BAR) domain [13]. Several in vitro experiments demonstrated that BAR proteins interact with synthetic lipid vesicles, deforming their membrane into tubules [13,14,15]. Such membrane-sculpting activities are attributed to a ~300 residue-long *N*-terminal domain conserved BAR domain. Apart from BAR proteins, the glycocalyx (GC)—a dynamic layer of sugars, proteins and lipids covering the surface of the cells—may also affect the formation of vesicular/tubular carriers for transport across endothelial cells. In endothelial cells, the GC covers the apical/luminal surface of the cell, and its dynamic and unique composition comprises proteoglycans, glycoproteins, glycosaminoglycans (GAGs) and adherent plasma proteins [16]. Although the GC is not featured in canonical models of membrane shape regulation, GC polymers can generate forces that deform the cell membrane [17]. Even though the BAR proteins and GC are abundant in the brain endothelium [7,18,19,20] and intrinsically associated with the cell membrane and membrane curvature, these elements are often neglected in terms of their role in the formation of vesicular or tubular carriers for the transport and intracellular fate of the cargo in endothelial cells.

Here, we review two elements that remain unexplored in transcytosis at the brain endothelium—BAR proteins and the GC—and attempt to raise the relevance of both in the mechanisms governing vesicular and tubular trafficking in transendothelial transcytosis.

## 2. BAR Proteins

The transcellular transport across the brain endothelium involves the deformation of cell membranes and recruitment of specialised proteins for the formation of vesicular/tubular carriers. Though it is generally accepted that the vesicular endocytic machinery mediates the transport in endothelial cells, it still remains to fully identify elements that affect the shaping of the cell membrane and guide the cargo from one side of the membrane to the other. A BAR domain is composed of three anti-parallel coiled-coil helices that allow the BAR proteins to self-polymerise into dimers with a crescent-shaped surface [13,21]. At this curved surface of the BAR domain, 12 cationic residues strongly interact with the anionic headgroups of the membrane lipids such as phosphatidylserine (PS) and phosphoinositides (PIPs) [22,23,24], therefore inducing membrane curvature. BAR domains differ in length, magnitude of intrinsic curvature and binding affinity to the cell membrane [22] and as such, these are commonly classified into three subgroups based on structural properties: N-BAR, FER-CIP4-homology-BAR (F-BAR) and inverse BAR (I-BAR) [25]. N-BAR and F-BAR proteins induce positive membrane curvature (i.e., the membrane bends in the direction of the leaflet decorated by the protein forming invaginations), while I-BAR triggers negative curvature (i.e., protrusions). Although the N- and F-BAR proteins induce deformation of cell membranes to form tubulations, the radius of these tubules is, generally, correlated with their shape. The F-BAR domain helices are relatively long and shallow (20 to 28 nm in diameter versus 60 nm [26]), favouring the deformation of larger volumes. When analysed in living cells, an N-BAR domain protein generates tubules from 20 to 60 nm wide, while the F-BAR tubules are 3-fold wider, reaching diameters of 60–100 nm [27]. Apart from the BAR domain, BAR proteins contain other variable domains targeting distinct cellular components with different moieties to guide their cellular location [22,28] (Table 1). Within the numerous members of BAR proteins, N-BAR (amphiphysin and endophilin) [18,29] and F-BAR proteins, including PACSIN-1 and PACSIN-2 [7,19], are expressed in the mammalian brain. These proteins have been implicated in intracellular trafficking in neurons; yet their actual role in BECs remains to be explored.

## 3. Glycocalyx

The glycocalyx (GC) is a layer of glycolipids and glycoproteins ubiquitously expressed on most cells and highly expressed on most endothelia, where it is highly involved in their homeostasis and integrity [20,30]. Glycoproteins are characterised by several carbohydrate chains (or glycans) covalently linked to a core protein, in which the carbohydrate may be in the form of a monosaccharide, a disaccharide, linear and branched oligosaccharides and a polysaccharide or their derivatives (such as a sulfonated or phosphorylated glycan). Proteoglycans are a subclass of glycoproteins, in which the carbohydrate units are polysaccharides containing GAGs [16,31]. GAGs are linear polymers of repeating disaccharide units comprising a hexosamine (N-acetylglucosamine and N-acetylgalactosamine) alternating with a uronic acid (glucuronic and iduronic acid) or neutral sugar (galactose). Examples include heparin, heparan sulphate (HS), keratan sulphate (KS), chondroitin sulphate (CS), dermatan sulphate (DS) and the non-sulphated anionic polysaccharide hyaluronic acid (HA) [32,33]. Except for HA, all other GAG repeating units undergo different levels of sulfation, bestowing them with bulky and negatively charged sulphate groups that confer rigidity and a high polarity to the polymer chain. In the particular case of BECs, HA is the most abundant GAG regulating monocyte migration and integrity of the endothelium [34,35]. Proteoglycans are a heterogeneous family composed of 43 members differing in their core protein, as well as the nature and number of GAGs attached to the core protein [31]. Recently, Iozzo et al. suggested a classification of proteoglycans into four classes: (1) intracellular secretory granules, (2) cell surface proteoglycans, which include transmembrane and glycosylphosphatidylinositol (GPI)-anchored proteoglycans, (3) pericellular basement membrane zone proteoglycans and (4) extracellular proteoglycans that are classified as hyalectan-lectincan (HA-binding and lectin-binding), spock and small leucin-rich proteoglycans [31]. Despite the diversity of proteoglycans, only specific ones are present in the endothelial GC. The cell surface proteoglycans, syndecans and glypicans, are of interest in the mechanisms of endocytosis in the endothelium and will be the main focus in this review (Figure 1).

There are four distinguished members of syndecans (-1, -2, -3 and -4) with a molecular size between 20 and 45 kDa and a short cytoplasmatic domain of ~40 amino acid residues, comprising a C1 and C2 region (proximal and distal to the cell membrane, respectively) separated by a variable region V, specific for each syndecan [36]. Each syndecan contains 3 to 5 GAG chains, in which HS is preferentially attached close to the N-terminus at the distal side of the cell membrane, while CS chains are located closer to the membrane surface. According to Song et al. [37], syndecan-2, -3 and -4 are present in the human and mouse brain endothelium, while syndecan-1 is not detected (Table 2). Apart from the brain endothelium, other endothelial cells express syndecans. The GC in the human lung microvasculature was confirmed by transmission electron microscopy, in which syndecans were described as essential proteins for the structure of the pulmonary endothelial barrier [38]. Human umbilical vein endothelial cells (HUVECs) have higher expression of syndecan-3 and -4 as compared with syndecan-1 and -2, contrary to the brain endothelium [39]. Glypicans are attached to membranes by GPI anchors with one or more HS linked to the core protein close to the cell membrane [31]. Within glypicans, there are six members (-1, -2, -3, -4, -5 and -6) with an average size of 60 kDa [40]. In humans, all glypicans are found in the brain endothelium (Table 2). Glypican-1 is associated with a protective role in the brain endothelium, and its loss is linked to dysfunctions in blood vessels during ageing [41]. In fact, the GC seems to be distinct depending on the tissue with an abundant expression of glypicans in the brain endothelium compared to that in the brain (Table 2). Apart from these cell surface proteoglycans, in the pericellular basement membrane, agrin is also abundant in the brain endothelium (Table 2). Interestingly, all agrin isoforms (with the exception of type II membrane agrin (TM-agrin)) are soluble glycosylated proteoglycans (~210 kDa) that form clusters with receptors, including low-density lipoprotein receptor-related protein 4 (LRP4) [42] and acetylcholine receptors [43]. TM-agrin is a transmembrane form of agrin abundant in axons and dendrites, in which it acts as a receptor or co-receptor [44] which has been implicated in the formation of filopodia-like protrusions in hippocampal neurons and fibroblast cells [45,46].

## 4. Transcytosis at the Brain Endothelium: Possible Role for BAR and GC?

The central nervous system (CNS) functions are maintained by a meticulous coordination of the activity of the multiple cells within the neurovascular unit (NVU), including vascular cells (endothelial cells, pericytes and smooth muscle cells), glia (astrocytes, microglia and oligodendrocytes) and neurons [4,5]. Within the NVU, BECs lining the brain blood vessels form the blood–brain barrier (BBB), which regulates molecular the flow between the blood and brain and is critical for appropriate functions of the brain (Figure 1). In the BBB, a continuous non-fenestrated brain endothelium is characterised by tight junctions (TJs), including claudin-3, -5 and -12 and occludin, and adherens junctions (AJs), which limit the crossing of molecules and ions [5,47,48]. Furthermore, BECs have a lower rate of transcellular vesicular transport than peripheral endothelial cells [49]. Other cells within the NVU further contribute to the properties of the BBB. Pericytes cover 60–70% of the abluminal BEC surface, while astrocytes endfeet reach up to ~ 99% of the surface overlapping pericytes [47,50]. Considering the differences in barrier properties between the brain and peripheral endothelium, Ben-Zvi et al. [51] compared the brain endothelium to the lung endothelium to identify the genes that specifically contribute to these unique properties of BECs. One of the genes enriched in BECs is the major facilitator superfamily domain-containing 2A (Mfsd2a). Indeed, genetic ablation of Mfsd2a resulted in a leaky BBB from the embryonic stages through to adulthood, while normal patterning of vascular networks is maintained. Electron microscopy demonstrated a dramatic increase in the transcellular vesicular transport in Mfsd2a-deficient (Mfsd2a^−/−^) mice without striking defects at the TJ levels [51]. These findings identified Mfsd2a as a regulator of the endothelium properties by suppressing the transcellular transport across BECs. Thus, these findings highlight how the distinct brain endothelium is compared to the peripheral endothelium. However, BECs express substrate-specific transport systems that facilitate influx and/or efflux of nutrients and regulatory molecules into and out of the brain, including transferrin receptor, low-density lipoprotein receptor-related protein 1 (LRP1) and receptor for advanced glycosylation end products (RAGE) [4,5]. Interestingly, Mfsd2a was also identified as the major transporter for docosahexaenoic acid (DHA), an omega-3 fatty acid essential for normal brain growth and cognitive functions, which is enriched in brain phospholipids. Lipidomic analysis revealed that Mfsd2a^−/−^ mice possess markedly reduced levels of DHA in the brain accompanied by a neuronal cell loss in the hippocampus and cerebellum, as well as cognitive deficits and severe anxiety [52]. Recently, it was also demonstrated that lipids transported by Mfsd2a mediate the suppression of a transcytotic route, which in turn ensures BBB integrity [53]. By using a transporter-dead Mfsd2a knock in mouse model as an in vivo tool, it was shown that Mfsd2a-mediated transport of DHA controls the luminal plasma membrane DHA lipid composition of the brain endothelium such that it actively prevents the formation of functional caveolae membrane domains. Thus, unlike the lung endothelium, which has low levels of DHA and possesses high levels of transcytotic caveolae vesicles, the Mfsd2a-mediated lipid transport in BECs renders them less capable of forming caveolae vesicles to act as transcytotic carriers, hence ensuring the integrity of the BBB [53].

The transcellular transport of molecules through trafficking membrane-enveloped carriers is defined as transcytosis [48,54,55]. Broadly, transcytosis comprises three steps: endocytosis, intracellular trafficking and exocytosis (Figure 1). Transcytosis was investigated in detail in the epithelium [56] and peripheral endothelium [57], and it is generally accepted that intracellular trafficking is regulated by the endo/lysosomal network. However, as recently demonstrated by Gu and colleagues [51,53], the brain endothelium presents distinct properties to the peripheral endothelium, and thus the cellular machinery for transcytosis in BECs might differ from the endothelial cells in other peripheral tissues. At BECs, transcytosis is mediated by the substrate-specific transport systems via receptor-mediated transcytosis (RMT) (Table 3) [4]. 

Interestingly, a few recent studies provided valuable detailed information about the expression of proteins responsible for transcellular transport across BECs in a mouse brain [58,59]. Once the cognate ligand binds to the specific receptor, transcytosis is initiated by endocytosis of the ligand–receptor complex, followed by trafficking across the cell to the opposite membrane and, ultimately, exocytosis. Intracellular trafficking through endosomes determines the fate of the vesicles filled with the receptor–ligand complexes, which will end up either in lysosomes for degradation or transported across by fusion with the other side of the membrane of BECs. The itinerary of intracellular trafficking is highly coordinated involving sorting by numerous regulatory proteins. However, what determines whether a cargo undergoes degradation or transcytosis in BECs is still to be identified. Notwithstanding this open question, RMT has been described in BECs for specific receptors (Table 3).

### 4.1. Endocytosis

Receptor-mediated endocytosis generates small (60–200 nm) membrane vesicles that transport various cargo molecules from the apical or basolateral membrane to inside the cell [60,61]. The cargo consists of transmembrane proteins (receptors) and their extracellular ligands. These cargos are involved in a broad range of physiological processes, including nutrient uptake, cell signaling and cell adhesion. Endocytic pathways in endothelial cells fall into two categories: clathrin-mediated endocytosis (CME) and clathrin-independent endocytosis (CIE) [62]. Conceptually, CME is a well-studied process that consists of a few sequential and partially overlapping steps [61]. CME is initiated by the clustering of endocytic coat proteins (such as clathrin) on the inner leaflet of the membrane, which is further continued by the recruitment of other coat proteins, including clathrin-adaptor proteins (heterotetrameric adaptor protein A2 complex, phosphatidylinositol-binding clathrin assembly protein (PICALM) and epsins). This assembly of coat proteins promotes membrane bending, which transforms flat membranes into “clathrin-coated pits” (CCPs). Following the recruitment of the coat proteins and membrane bending, dynamin and BAR-containing proteins (such as amphiphysin and endophilin) constrain and cut the neck of the invagination to separate the clathrin-coated vesicles from the membrane (Table 4). Lastly, the cargo-filled vesicles are released for further intracellular trafficking in endosomes within the cell [61]. This mechanism of CME is the most well characterised, and few studies have identified ~20 different receptors at the surface of BECs that are associated with CME for transcytosis across the brain endothelium [63].

An N-BAR protein, amphiphysin, is found in two isoforms, -1 and -2, with both containing an Src homology 3 (SH3) domain that interacts with dynamin [64] and N-WASP [65] as well as a central region containing a CLAP domain, which mediates the binding to clathrin and the clathrin-adaptor protein (AP2) [66,67], suggesting a role in CME. Experimental manipulations of living cells showed that disruption of the interaction of amphiphysin and dynamin inhibited CME [68]. These studies suggested that amphiphysin contributes to dynamic curvature formation at the neck of the budding endocytic vesicles during CME by coordinating with other curvature-inducing proteins (dynamin). Similar to amphiphysins, endophilins interact with dynamin and N-WASP through a SH3 domain [69], and thus endophilin is also implicated in a fission step in CME [70,71]. Other than N-BAR, F-BAR proteins have also been demonstrated to be involved in CME. Within the family of F-BAR proteins, CIP4s, there are three CIP4-like proteins: CIP4, forming-binding protein 17 (FBP17) and transactivator of cytoskeletal assembly-1 (Toca-1). CIP4 and FBP17 participate in the initiation and scission of endocytic vesicles in CME [72,73]. The F-BAR FCHO subfamily includes two members, FCHO1 and FCHO2, containing an N-terminal F-BAR and C-terminal mu-HD domain. FCHO1/2 bind to the cell membrane via the F-BAR domain and recruit binding partners Eps15 and intersectin that, in turn, engage AP2 to initiate CME. FCHO1/2 accumulate at CCPs at an early stage and are recruited prior to other BAR proteins, such as FBP17 [74]. Complete loss of CCPs is observed when FCHO1/2 levels are greatly reduced with a concomitant reduction in uptake of three cargos for CME, including transferrin, low-density lipoprotein and epidermal growth factor [74,75]. The FCHO1/2 role in CME has been shown in fibroblasts, astrocytes and neurons; however, there is still no evidence in the endothelium. Another F-BAR family, FCHSDs, includes two members: FCHSD1 and FCHSD2. Each FCHSD contains an F-BAR domain at the N-terminal and two SH3 domains at the C-terminal. FCHSD1/2 interact with WASP through one SH3 domain, triggering the Arp2/3 complex to nucleate actin filaments. Recent studies [76,77,78] have revealed the function of FCHSD2 in CME. Almeida-Souza and colleagues [78] suggested a model for FCHSD2 function, in which, after CME initiation, FCHSD2 is recruited to CCPs by intersectin via an SH3–SH3 interaction, and intersectin-FCHSD2 complexes accumulate at the edge of the CCPs. The FCHSD2 F-BAR domain and the presence of late PI(3,4)P2 drive the binding of FCHSD2 to the planar region opposed to CCPs, allowing it to activate actin polymerisation. The FCHSD2-dependent actin structure contributes to invagination and maturation of CCPs. In fact, FCHSD2 deletion leads to reduced transferrin uptake [78]. Furthermore, FCHSD2 also contributes to CCP initiation and EGFR endocytic trafficking [77].

Within the brain endothelium, despite the prevalence of CME, CIE mechanisms may also regulate endocytosis. Among the CIE mechanisms, caveloae-mediated vesicle trafficking consisting of flask-shaped pits coated with caveolin-1 predominates in peripheral endothelial cells [85,86]. Recent studies in BECs showed that caveolae-mediated endocytosis is inhibited by Mfsd2a to maintain the integrity of the BBB [51,53]. With ageing, the BBB has been shown to be leakier [87,88]. Indeed, in a recent study, a shift from ligand-specific RMT to non-specific caveolar transcytosis with ageing was exhibited [88]. Most genes encoding for putative RMT receptors, including *Tfrc*, declined with age, as well as proteins involved in CME, including clathrin, PICALM, Rin3 and Epsin15. Interestingly, expression of *Cav1*, which encodes for caveolin-1, was amplified with age along with a decline in *Mfsd2a*, the suppressor of the formation of caveolae at BECs [88]. Further CIE types, including ADP-ribosylation factor 6 (Arf6)-mediated endocytosis, clathrin-independent carriers/glycosylphosphatidylinositol-anchored protein-enriched endocytic compartments and endophilin-mediated endocytosis, have been reported in non-endothelial cells [89]. Arf6 belongs to the ARF family of six small guanosine triphosphate (GTP)ases, and it has been implicated in CME and CIE mechanisms [90,91]. Arf6 and its exchange factor for Arf6 (EFA6) have been shown to be involved in transferrin receptor-mediated transcytosis, in which the N-BAR-containing protein endophilin cooperates with EFA6 to activate Arf6 and regulate CME [92]. Yet a clathrin-independent Arf6-mediated pathway has also been described, in which tubular rather than vesicular structures were found [93]. Boucrot et al. [79] provided evidence for a fast endophilin-mediated endocytosis (FEME). FEME is clathrin- and AP2-independent and dynamin-dependent and operates from distinct cell regions on a different timescale to CME. Therefore, FEME is defined as the rapid formation (within seconds) of endophilin-positive tubulo-vesicular (<1 µm) carriers at the surface of the cell that rapidly travel towards the center of the cell upon stimulation of the cargo receptor by the cognate ligand [79]. FEME mediates the ligand-triggered endocytosis of several G protein-couple receptors (α2*a*- and β1-adrenegic, dopaminergic D3 and D4 receptors), receptor tyrosine kinases (EGFR, HGFR, VEGFR, PDGFR, NGFR, IGF1R) and interleukin-2 receptors [79]. Furthermore, FEME mediates the cellular entry of established clathrin-independent cargos such as cholera and shiga toxin [80]. Although endophilin is implicated in FEME, the mechanism of tubulation has not yet been fully elucidated. Simunovic et al. [94] shed light on the scission of the tubular structures through a mechanism of friction-driven scission, in which BAR proteins coating the membrane tubulation create a frictional barrier on the tubule that leads to the scission through lysis. Despite the significance of FEME, these studies are focused on the epithelium and there is a need to further explore these mechanisms in trafficking across the brain endothelium.

Finally, we recently revealed that the F-BAR protein PACSIN-2 (or syndapin-2) is of particular interest for transcytosis in BECs [7]. PACSIN-2 is involved in CME [95,96], biogenesis of caveolae and caveolae-mediated endocytosis [81,82], clathrin- and caveolae-independent mechanisms of endocytosis [7,83]. Although PACSIN-2 has been implicated in CME, evidence is needed to decipher its actual role. In fact, PACSIN-2 has recently been more associated with CIE. Senju et al. [81] reported that PACSIN-2 mediates the shape formation of caveolae by direct interaction with caveolin-1 and that knockout of PACSIN-2 results in an abnormal morphology of the caveolin-1-associated membranes. PACSIN-2 also recruits dynamin-2 for the vesicle fission in caveolae-dependent endocytosis [81,82]. Apart from the CME and CIE mechanisms, Chandrasekaran et al. [83] described that the cellular uptake of Clostridium difficile toxin A (TcdA) is mediated by PACSIN-2 through a clathrin- and caveolae-independent pathway in epithelial cells. TcdA toxin showed colocalisation with PACSIN-2-positive structures and disruption of PACSIN-2 function inhibited TcdA uptake and toxin-induced downstream effects. Based on these findings, it was thus hypothesised that PACSIN-2 mediates vesicle or tubule formation for endocytosis. Interestingly, our recent study demonstrated that PACSIN-2 participates in the formation of tubular carriers for a fast transcytosis of LRP1 across BECs [7]. Furthermore, we also demonstrated that BECs employ PACSIN-2 for the clearance of amyloid-ß via LRP1 across the BBB [84]. Hence, our findings corroborate the hypothesis that PACSIN-2 facilitates a tubulation-mediated mechanism for transcytosis across BECs. Meanwhile, the full machinery remains to be elucidated.

In regard to the involvement of the GC in endocytosis in the brain endothelium, the concept remains elusive. Nevertheless, some evidence has highlighted the interaction of ligands with the negatively charged GC brush at the surface of BECs and whether those interactions contribute to their transport across the endothelium. In particular, proteoglycans [97] and GAGs, including HS and HA [98], have been connected with the transport of several molecules by acting as receptors that mediate endocytosis. In terms of proteoglycans, syndecans are involved in the uptake of exosomes [99]. In addition, syndecan-4 acts as an endocytic co-receptor by interacting with the fibroblast growth factor receptor 1 (FGFR1) for FGF2 uptake through a lipid raft-dependent, clathrin- and dynamin-independent manner through activation of small GTAse Rac1 [100]. In human cerebral microvascular endothelial cells (hCMEC/D3), the treatment with heparinase III resulted in a reduction in syndecan-2 levels and consequently a decrease in the exosome uptake, which indicates a role for syndecans as receptors for exosomes in BECs [99]. Glypicans are also implicated in endocytosis. For instance, it has been proposed that LRP1 forms a complex with glypican-1/PrP^c^ to control amyloid-β endocytosis via CME [101]. In neuronal cells, HS cooperates with LRP1 to mediate cellular uptake of amyloid-β [102], while in fibroblasts, Sonic hedgehog (Shh) internalisation occurs due to interaction between HA chains with glypican-3 and LRP1 [103]. The type II transmembrane glycoprotein hepatic asialoglycoprotein receptor was the first lectin identified, and it has been a classical system of RMT [104]. Another endocytic lectin, hyaluronan receptor for endocytosis (HARE), is associated with endocytosis of GAGs, such as HA and heparin, mediating their endocytic clearance from the vascular and lymphatic systems [105,106]. Although HARE is present in ependymal cells lining the ventricles in the brain, its presence and role in BECs have not been described.

Besides the role of GAGs anchored to proteoglycans, their role has been emphasised in the entry of virus [32,107,108]. Adeno-associated virus (AAV) has been used as a vector for delivery into the brain by introducing peptides on the capsid surface of the virus. Geoghegan et al. compared the binding preference of an engineered peptide-modified AAV serotype 2 (AAV-GMN) that displays a heptamer peptide, GMNAFRA, for in vivo brain vascular targeting, to the unmodified AAV2 vector [109]. In this study, it was described that the AAV-GMN vector, unlike AAV2, uses chondroitin sulphate as its primary cellular response. Interestingly, it was also found that while both AAV-GMN and AAV2 can bind to heparin, only AAV2 efficiently uses HS as a functional receptor for transduction. This finding provides an insight into the transduction of AAV2 and indicates that targeting chondroitin sulphate, but not heparan sulphate, is an effective strategy for in vivo targeting. Moreover, it has also been described that clusterisation of GAGs is associated with internalisation of cell-penetrating peptides by creating rearrangement of actin that, ultimately, leads to endocytosis [110,111].

### 4.2. Intracellular Trafficking: Spherical versus Tubular Carriers

Once a ligand–receptor complex is internalised via CME or CIE, the cargo is sorted by the common endosomal sorting network that determines its fate—either degradation by trafficking into lysosomes, recycling back to the cell membrane or transcytosis by fusion with the opposite side of the membrane. The classical sorting into vesicular carriers—endosomes—is present in many cell types across tissues [112,113,114], and BECs likely employ similar processes of endosomal maturation for intracellular trafficking. The vesicular endosomal pathway is a spatiotemporal succession of distinctive compartments, which continuously interchange their content while undergoing structural transformations [115,116]. A generally established concept is that endosomes undergo maturation, starting from the initial early endosomes (within 1–5 min, vesicles positive for Rab5 and early endosome antigen 1, EEA1), late endosomes (10–15 min, Rab7-positive vesicles) and lysosomes for degradation (after 30 min of the initiation of CME in vesicles associated with a lysosomal-associated membrane protein 1, LAMP1) [115]. However, this timing (15 to 30 min) is inconsistent with the rapid rate of transcytosis observed in vivo, which has been found to occur in <60 s in the lung epithelium [85] and even faster in BECs [7,117].

Mounting evidence from the early 1970s to the 1990s proposes intracellular trafficking through a network of tubules in the brain endothelium [8,9,10,11,12]. Bundgaard and colleagues performed electron microscopy (EM) of hagfish BECs. Three-dimensional reconstructions of serial EM sections showed that intracellular membranes arising from transcytosis were rarely single vesicles, but instead part of a large multidimensional network of tubules [11,12]. Recent studies shed light on the tubulation-mediated transcytosis of transferrin [6] and LRP1 [7] across BECs (Figure 2). In both studies, it was shown that the avidity of the ligands controls the intracellular trafficking of the ligand–receptor complex across BECs and that trafficking via tubules facilitates a fast shuttling across BECs, avoiding lysosomal degradation. Therefore, these studies suggest that trafficking through tubules is reclaiming a role of paramount importance for the shuttling of molecules across the BBB.

At the surface of BECs, the GC may act as an endocytic receptor or affect the interaction of ligands with their cognate receptors. However, pieces of evidence are available for the implication of the GC in the trafficking of the cargo through the endothelium, particularly in terms of vesicular/tubular carriers. Interestingly, the GC role in the formation of membrane projections, tubular extensions and protrusions has been found in the epithelium [17]. The endothelial GC assembles into polarised long chains, creating apical membrane vesicles. Furthermore, it was shown that the vesicles containing GC members elongate into vacuole-like structures along the length of the cell to be exposed on the surface of endothelial cells [118,119]. However, the role of the GC in membrane tubule formation for endocytosis is poorly understood. The importance of the GC during cell migration through endothelial cells was suggested in a cancer model. Circulating tumour cells produce an adhesive vascular niche by the accumulation of HA shed on the endothelial GC. This allows the transendothelial migration and invasion of tumours into the stroma through the binding of the CD44v glycoprotein to components of the subendothelial extracellular matrix [120].

### 4.3. Exocytosis

Exocytosis consists in fusion of the intracellular vesicle and/or tubule with the plasma membrane. By studying synaptic vesicles, Söllner et al. [122] identified three main components essential for membrane fusion in exocytosis: *N*-ethylmaleimide soluble factor (NSF), synaptosomal-associated proteins (SNAPs) and SNAP receptors (SNAREs). SNAREs are divided into two categories: vesicles (*v*-SNARE) that are incorporated into the membrane of the intracellular vesicle and targets (*t*-SNARE), which are associated with the opposing target membrane [123]. All SNAREs contain the characteristic SNARE motif of ~70 amino acids comprising heptad repeats that acts either on *v*-SNAREs or *t*-SNAREs. Once a vesicle is in close contact with the cell membrane, the interaction between the *v*- and *t*-SNAREs leads to the formation of a *trans*-SNARE complex (or SNARE-pin), in which four SNARE motifs assemble as a twisted parallel four-helix bundle that facilitates the membrane fusion [124]. Following fusion, the remaining SNARE complex on the fused membrane is referred to as a *cis*-SNARE complex, which then undergoes disassembly by NSF and SNAP to recycle the SNAREs for other fusion events [124]. Though SNAREs are well characterised in synaptic vesicles, the involvement of SNAREs in transcytosis and, in particular, transcytosis in BECs is far less understood [125].

Despite the recent progress in the understanding of the GC structure and role in endocytosis, scarce studies have been focused on the GC in exocytosis. Yet it has been described that HS shedding in endothelial cells is associated with a differential rate of exocytosis turnover [126].

## 5. Conclusions and Future Remarks

The brain endothelium, lining the brain blood vessels, triggers the restrictive properties of the BBB, with specialised TJs preventing paracellular transport from blood-to-brain, and unusually low levels of vesicular trafficking, limiting transcellular passage or transcytosis. Nonetheless, BECs employ multiple substrate-specific transport systems to mediate transport of essential molecules by RMT. Different receptors have been identified to undergo transcytosis; however, it yet remains to be clarified how BECs differentiate between transcytosis or degradation, as well as how the full molecular machinery is implicated in transcytosis. What determines whether a cargo undergoes transcytosis or degradation in BECs? What molecular machinery is involved in the shuttling of a cargo across BECs?

Transcytosis involves the deformation of membranes and recruitment of specialised proteins for the formation of vesicular and/or tubular endocytic carriers. Surprisingly, even though BAR proteins and the GC are intrinsically associated with cell membranes and known to control the shape of the membrane, their role is often neglected in the brain endothelium. Numerous BAR proteins are enriched in a mammalian brain, and particular BAR proteins have been reported to be altered in CNS disorders, such as Alzheimer’s [29,84] and Huntington’s [127,128] disease, and epileptic seizures [129]. Thus, it appears that BAR proteins may have an important role in the brain that is worth further exploring. Similarly, although the GC covers a large surface area of BECs, there is scarce evidence of its the role in transcytosis across the BBB. Considering that the GC brush is a main selective barrier for the entrance of molecules into the brain, it seems evident that the GC has a role in transcytosis across the brain endothelium. However, due to the complexity of the GC and difficulties in determining the precise structure of the GC, its role in transcytosis is yet to be fully understood. The thickness of the GC remains debatable, especially in humans, as well as whether it is altered in pathological conditions, such as tumours, neurodegeneration and ageing. Most of the techniques used to visualise the GC thickness are based on label markers, such as wheat agglutinin lectin, which is only specific for a few GC components, including heparan sulphate and hyaluronan. Further studies or visualisation methods are required to circumvent this limitation. The GC composition and complexity indicates that this brush covering BECs has more functions than homeostasis and maintenance of the integrity of the BBB. Recent evidence shed light on the relevance of GC polymers, such as glycoproteins and GAGs, for the curvature of membranes and endocytosis [17,97,104]. The involvement of other GC polymers during transcytosis in BECs is yet to be unveiled. However, could the GC polymer brush affect the endocytosis of different receptors and control the intracellular trafficking? GC polymers are capable of deforming the membrane into tubular structures forming protrusions. Could the GC brush contribute to the shape of the intracellular vesicular or tubular endocytic carriers?

A deeper understanding of the mechanisms regulating transcytosis in the brain will not only set the stage for exciting opportunities of studies to comprehend CNS disorders but will also reveal new strategies for CNS delivery. Indeed, numerous elements intrinsically associated with cell membranes remain unexplored even though they could be critical pieces of the mechanism of transcytosis.

## Figures and Tables

**Figure 1 cells-09-02685-f001:**
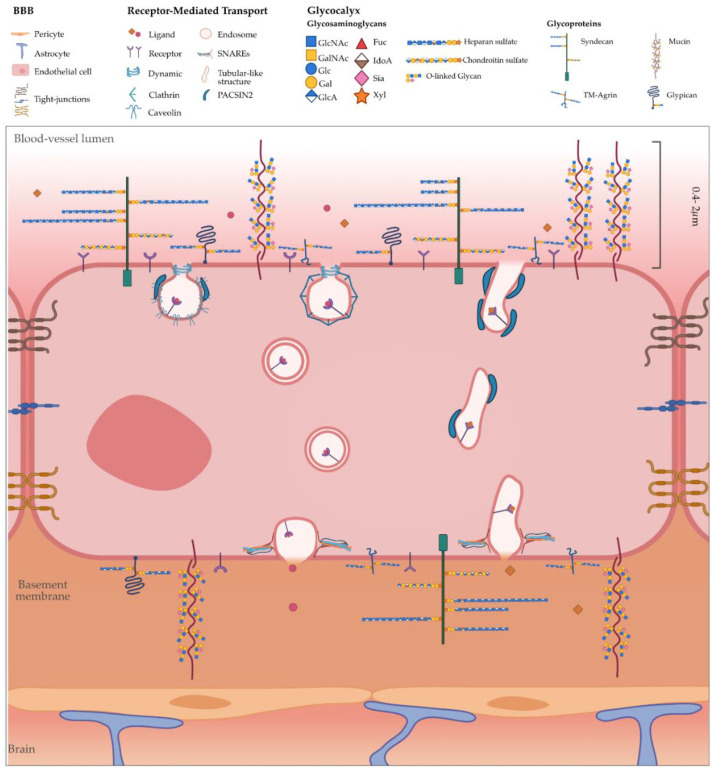
Schematic representation of the glycocalyx (GC) in the blood–brain barrier (BBB) and the mechanisms of transcytosis. The BBB consists of vascular cells (endothelial cells, pericytes and smooth muscle cells), glia (astrocytes, microglia and oligodendrocytes) and neurons. The non-fenestrated brain endothelium is characterised by the presence of tight junctions. Moreover, a layer of the GC covers the luminal/apical surface of the endothelium (0.2–2 µm). The GC brush is highly dynamic with a unique composition of glycoproteins and glycans. Within the glycoproteins, proteoglycans, such as syndecans, glypicans and TM-agrin, contain GAGs (heparan and chondroitin sulphate chains) anchored to the core protein. These proteins may affect the binding of ligands to their receptors and endocytosis. Once the ligand binds to the receptor, the cargo undergoes endocytosis via caveolae- or clathrin-dependent or clathrin-independent mechanisms, with intracellular trafficking mediated by endosomal/lysosomal or a direct route mediated by tubular carriers, and eventually exocytosis.

**Figure 2 cells-09-02685-f002:**
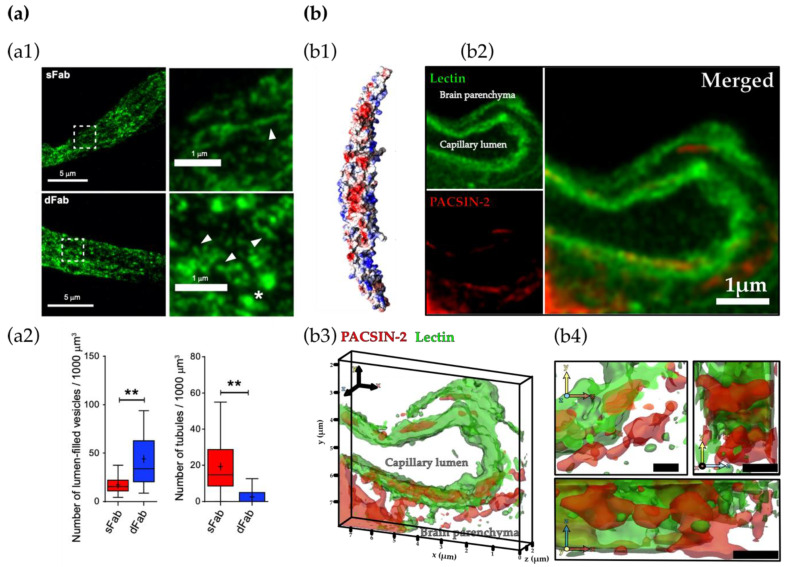
Tubulation as a mechanism of transport across the brain endothelium. (**a**) Transport of transferrin receptor-based brain shuttle constructs that are engineered by fusing a single-chain Fab fragment of an anti-transferrin monoclonal antibody to either one (monovalent, sFab) or both C-terminal ends (bivalent, dFab) across the brain endothelium. (**a1**) Representative images of mouse brain capillary with brain shuttle platform based on sFab or dFab (green). Arrowheads point to an individual tubule for sFab and short elongated buds for dFab. Star shows vesicles with a dFab signal within the lumen. (**a2**) Number of lumen-filled vesicles and tubules for sFab and sFab in segments of brain capillaries revealing the differential transport along intracellular tubules in vivo. Permission from [6]. ** represents the statistical significance between sFab and dFab. (**b**) Tubulation in mouse brain endothelial cells (BECs) mediated by PACSIN-2. (**b1**) Structure of PACSIN-2. Permission from [121]. (**b2**) Section of brain capillary stained by lectin (in green) and PACSIN-2 (red) imaged by stimulated energy depletion (STED) microscopy and (**b3**) corresponding 3D rendering showed as projection and (**b4**) close-up from top, bottom and side view. Adapted from [7].

**Table 1 cells-09-02685-t001:** N-, F- and I-BAR family domain structure and cellular functions. Adapted from [28].

**BAR Proteins**	**Protein Domains**	**Cellular Functions**
**N-BAR**		
Amphiphysin-1	SH3	Endocytosis
Amphiphysin-2	-	
Endophilin-1, -2, -3	SH3	
**F-BAR**		
CIP4 (CIP4, FBP17, Toca-1)	SH3, WW	Endocytosis, Phagocytosis (FBP17), Filopodium (Toca-1), Lamellipodium (CIP4)
FCHO (FCHO-1, -2)	Mu-HD	Endocytosis
SrGAP (SRGAP-1, -2, -3, -4)	RhoGAP, SH3	Filopodium
PACSIN (PACSIN-1, -2, -3)	Tyr-kinase, SH3	Endocytosis, Filopodium (PACSIN-2)
PSTPIP (PSTPIP-1, -2)	SH3	Endocytosis, Filopodium (-2), Lamellipodium (-1)
FCHSD (FCHSD-1, -2)	SH3	Endocytosis
FES/FER	FX, SH2 Tyr-kinase	Lamellipodium
NOSTRIN	SH3	Endocytosis
GAS7	SH3, WW	Filopodium
**I-BAR**		
IRSp53, IRTKS, MIM	CRIB, SH3, WH2-like motif	Endocytosis, Filopodium, Lamellipodium

Abbreviations: BAR, Bin/Amphiphysin/Rvs; SH3, Src homoly-3; CIP4, CD42-interacting protein-4; FBP17, forming-binding protein 17; Toca-1, transactivator assembly-1; Mu-HD, Mu-homology domain; SrGAP, SLIT-ROBO Rho GTPase-activating protein; Rho-GAP, Rho GTPase-activating protein; Tyr-kinase, tyrosine-kinase; PSTPIP, proline-serine-threonine phosphatase-interacting protein; FX, F-BAR extension; SH2, Src-homology 2; CRIB, CD-42- and Rac-interactive binding.

**Table 2 cells-09-02685-t002:** Proteoglycans expression in human and mouse brains and brain endothelia. Adapted from [37].

**Proteoglycans**	**Human**		**Mouse**	
	**Brain Endothelium**	**Brain**	**Brain Endothelium**	**Brain**
Syndecan-1	−	−	−	−
Syndecan-2	++/+++	++	++	+/++
Syndecan-3	+++/++++	+++/++++	+++/++++	+++
Syndecan-4	++	+++	+/++	+
Glypican-1	+	+/++	+	+
Glypican-2	−/+	+	−	+
Glypican-3	+	−	+	−
Glypican-4	+/++	+	+	+
Glypican-5	+/++	++	++	+
Glypican-6	+	−	+	+
Agrin	+/++	+	+++	++

Abbreviations: (−) not detected (= 0); (+) low (<500); (++) medium (<1000); (+++) high (>1000); (++++) very high (>2000) expression in the tissue.

**Table 3 cells-09-02685-t003:** Receptors at the brain endothelium initiating receptor-mediated transcytosis.

**Receptor**	**Natural Ligands**	**Direction**
Transferrin (TfR)	Transferrin	Blood–Brain
Insulin (IR)	Insulin	Blood–Brain
Leptin (LepR)	Leptin	Blood–Brain
Low-density lipoprotein receptor related protein 1 (LRP1)	Lipoproteins, apolipoprotein E (ApoE), α_2_-macroglobulin, aprotinin, amyloid-ß	Blood–Brain Brain–Blood
Receptor for advanced glycosylation end products (RAGE)	Glycosylated end products, amyloid-ß	Blood–Brain

**Table 4 cells-09-02685-t004:** N- and F-BAR proteins involved in transcytosis steps: endocytosis and trafficking.

**BAR**	**Function**	**Mechanism**	**Receptor**	**Tissue**	**Ref.**
**N-BAR**					
Amphiphysin	Vesicle initiation and fission	CME	TfR	Fibroblasts	[64,68]
Endophilin	Vesicle initiation and fission	CME	-	-	[70,71]
	Tubulo-vesicular carriers	CIE/FEME	GPCRs, TKs, IL-2	Epithelium	[79,80]
**F-BAR**					
CIP4, FBP17	Vesicle initiation and fission	CME	GLUT-4	-	[72,73]
	Priming of FEME	CIE/FEME			
FCHO1/2	Formation of CCPs (early stages of CME)	CME	TfR, LDLR, EGFR	Fibroblasts, neurons, astrocytes	[74,75]
FCHSD2	Initiation, invagination and maturation of CCPs	CME	TfR EGFR		[76,77,78]
PACSIN-2	Caveolae biogenesis	CIE	-	-	[81,82]
	Tubulo-vesicular carriers	CME/CIE?	LRP1	Brain endothelium	[7,83,84]

Abbreviations: CME, clathrin-mediated endocytosis; CIE, clathrin-independent endocytosis; TfR, transferrin receptor; FEME, fast endophilin mediated endocytosis; GPCRs, G-protein coupled receptors; TKs, tyrosine kinase receptors; IL-2, interleukin-2; GLUT4, glucose transporter 4; CCPs, clathrin-coated pits; LDLR, low-density lipoprotein receptor; EGFR, epidermal growth factor receptor; LRP1, low-density lipoprotein receptor related protein 1.

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
