# Peer review of "The Role of BAR Proteins and the Glycocalyx in Brain Endothelium Transcytosis"

_cells, 2020, doi:10.3390/cells9122685_

Round 1
Reviewer 1 Report
The review nicely summarizes important concepts which deserve substantial attention in the field. This work is of interest to the readership of the journal Cells. The authors provide a stimulating review of important concepts in BBB transcytosis, including spherical versus tubular transcytosis invaginations, modification by glycocalyx, and modification by BAR proteins. This exciting manuscript would be improved by revision to Figure 1, as well as editing for grammar and clarity.
- Caveolar endocytosis and transcytosis should be included in the text as part of introduction of CIE and as the mechanism of MFSD2a suppression of transcytosis (eg line 183, 225), and in Figure 1, adjacent to the clathrin endocytosis, in association with PACSIN-2. Caveolar endocytosis is a substantial mechanism of receptor mediated endocytosis in the BBB. Note that Reference 89 elegantly shows an age-related increase in caveolar BBB trafficking, but does not include data to address whether or not caveolar BBB trafficking is receptor mediated or non-specific. This should be amended in the text.
- Figure 1: Several elements of the cartoon should be modified to reflect current understanding of the biology of the BBB more closely. For example, tight junctions are homotypic interactions between adjacent BECs, so where in the current drawing single molecules are drawn, instead it should be pairs of molecules with one set contributed by each BEC. Pericytes lie parallel to the BEC layer and do not extend processes past the astrocyte endfeet. Is there a way to graphically represent the mechanism(s) by which glycocalyx and BAR proteins influence transcytosis? Minor comment, the color of the astrocyte in the legend should more closely match the color of the astrocyte in the figure.
Minor comments
- In the section regarding impact of EC GC on exocytosis, are the authors highlighting the role of abluminal or luminal GC (i.e. to influence exocytosis exclusively in brain-to-blood directed or also blood-to-brain directed transcytosis)? No glycoprotein or extracellular matrix components are depicted in figure 1 on the abluminal aspect of the endothelial cells.
- The section on how GC modifies endocytosis could be better organized. For example, consider providing an overview statement describing the mechanisms by which GC components could impact endocytosis, transcytosis, and exocytosis. It might be worthwhile to add putative mechanistic information to one of the tables; for example, to indicate whether the impact of each GC component is by inducing membrane invaginations, engaging in low affinity charge-based or high-affinity ligand binding (eg syndecan:FGF), etc. The discussion of how GC modifies endocytosis (line 322) should follow directly after the description of the GC, and would benefit from having a dedicated subheading. Discussion of Mucin-1 inducing membrane invaginations (line 150-152) could be moved to the section describing how GC modifies endocytosis/transcytosis (line 322-). What is the quantitative expression level of mucins in epithelial cells versus brain endothelial cells?
- Line 166: BEC do not undergo endocytosis; cargo undergoes endocytosis.
- Line 159-160, 171-172: microglia are a subset of glia.
- Line 159-160, 171-172: oligodendrocytes are not commonly considered part of the BBB.
- Line 197-198 unclear
- Line 143 Table 2: In addition to citing the Human Protein Atlas, please also cite relevant primary literature describing the expression of proteoglycans in brain and non-brain endothelium.
- Table 2. It is not clear why the authors are comparing brain and colon. Is the purpose of this comparison is to show that the proteoglycans are broadly similar, with a few key differences (e.g. brain enrichment for Glypican 5-6), suggesting that Glypican 5-6 confers the suppression of transcytosis characteristic of the BBB relative to the colon? If yes, it would be helpful to specifically discuss how Glypican 5-6 influence transcytosis. If this does not reflect the authors’ intentions, greater clarification is warranted.
- Line 205: why compare the human brain endothelium to the tick epithelium?
- Table 2 and 4: please align the text across columns.
- Line 270 Table 4. Not clear why non-BBB cells (epithelium, fibroblasts) are included in this table, unless if adding discussion of BAR proteins in blood-CSF barrier or blood-meningeal barrier structures that contain epithelial cells and fibroblasts, respectively.
- Please minimize excessive abbreviations which reduce readability. For example, the abbreviation CLIC/GEEC is introduced line 287 but not used again.
- Line 296: Is there interplay between FEME and other CIE mechanisms, given the common deployment of some protein binding partners?
- Figure 2. Unclear what the arrowheads and asterisk in a1 point indicate. The figure legend should be expanded to better explain the important (and beautiful) structural features in these images.
- Line 232 typographical error: constrains, not constraints.
- Line 326. The phrase “endocytosis receptor” will be unfamiliar to a broad readership; please define.
- Line 328-331 contains two sentences that are unclear and appear to be contradictory.
- Line 344 spelling error: emphasized
Author Response
The review nicely summarizes important concepts which deserve substantial attention in the field. This work is of interest to the readership of the journal Cells. The authors provide a stimulating review of important concepts in BBB transcytosis, including spherical versus tubular transcytosis invaginations, modification by glycocalyx, and modification by BAR proteins. This exciting manuscript would be improved by revision to Figure 1, as well as editing for grammar and clarity.
We thank the reviewer very much for the feedback. We are pleased to address all reviewer’s comments. Please find below the response to each comment, and alterations highlighted in yellow in a revised version of the manuscript.
- Caveolar endocytosis and transcytosis should be included in the text as part of introduction of CIE and as the mechanism of MFSD2a suppression of transcytosis (e.g., line 183, 225), and in Figure 1, adjacent to the clathrin endocytosis, in association with PACSIN-2. Caveolar endocytosis is a substantial mechanism of receptor mediated endocytosis in the BBB. Note that Reference 89 elegantly shows an age-related increase in caveolar BBB trafficking, but does not include data to address whether or not caveolar BBB trafficking is receptor mediated or non-specific. This should be amended in the text.
We appreciate the careful reading of the manuscript. With regard to caveolar endocytosis, we recognise its significance, and so we have added it to the schematic in Figure 1. In the section 4, we start describing the blood-brain barrier and receptor-mediated transcytosis without going into detail about the specific mechanisms of transport across the brain endothelium. Thus, when describing Mfsd2a, we aim here to ascertain the difference between brain and peripheral endothelium (such as the lung endothelium). Yet, we have added the following sentence: “Recently, it was also demonstrated that lipids transported by Mfsd2a mediate the suppression of a transcytotic route, which in turn ensures BBB integrity” to initiate the role of Mfsd2a in caveolar endocytosis (line 194). Then, in the section 4.1, in which we introduce the endocytic pathways, we introduce caveolar endocytosis as CIE and give details about its suppression by Mfsd2a.
With regards to the age-related shift in transcytosis elegantly demonstrated by Yang et al. 2020 (ref 94), it is clearly stated a global shift from ligand specific RMT to non-specific caveolar transcytosis with age in brain endothelial cells. The authors demonstrated that most genes encoding putative RMT receptors, including Tfrc, decreased with age, as did downstream clathrin components and its adaptors. Expression of Cav1, which encodes the principal component of caveolae, trended upwards with age, whereas Mfsd2a, the product of which suppresses caveolae formation, decreased. Further, the authors assessed receptor-mediated and caveolar transport using their canonical ligands, endogenous and injected. With mass spectrometry, it was observed a decrease in endogenous, RMT-transported transferrin and a concomitant increase in caveolae-transported albumin within perfused, aged microvessels. This was also consistent with an age-related decline in TFRC and clathrin expression, and an increase in ligand non-specific caveolae. It was also observed similar patterns using the injected RMT ligand leptin and a caveolar-transported ligand, horseradish peroxidase, with young endothelium taking up more of the leptin but the aged endothelium taking up more of horseradish peroxidase. Hence, in terms of caveolar trafficking being receptor mediated or non-specific, the authors indicated that there is an increase in ligand non-specific caveolae based on their experiments with horseradish peroxidase. In our manuscript, we refer to the alterations in transcytosis with age based on these findings.
- Figure 1: Several elements of the cartoon should be modified to reflect current understanding of the biology of the BBB more closely. For example, tight junctions are homotypic interactions between adjacent BECs, so where in the current drawing single molecules are drawn, instead it should be pairs of molecules with one set contributed by each BEC. Pericytes lie parallel to the BEC layer and do not extend processes past the astrocyte endfeet. Is there a way to graphically represent the mechanism(s) by which glycocalyx and BAR proteins influence transcytosis? Minor comment, the color of the astrocyte in the legend should more closely match the color of the astrocyte in the figure.
We thank the reviewer for the valid suggestion. We have altered the schematic to better represent the structure of the tight junctions between adjacent BECs, the schematic symbol for the pericytes, and the colour of the astrocytes to ensure that colours are matching the legend. With regards to the graphic representation of the mechanism(s) by which glycocalyx and BAR proteins affect transcytosis, evidence of the role of these elements in transport across the BBB is diverse and focused on individual receptors, and in specialised tissues with little reference to the brain. Hence, it is complex to summarise the diverse information that remains to still define clear roles for BAR proteins and glycocalyx, particularly, in the BBB. We attempted with our review to highlight the vital role of these elements yet as we reinforced in our manuscript, there is still to be clearly elucidated their influence in transcytosis.
- In the section regarding impact of EC GC on exocytosis, are the authors highlighting the role of abluminal or luminal GC (i.e., to influence exocytosis exclusively in brain-to-blood directed or also blood-to-brain directed transcytosis)? No glycoprotein or extracellular matrix components are depicted in figure 1 on the abluminal aspect of the endothelial cells.
Regarding the role of glycocalyx in exocytosis, we are referring to luminal (brain-to-blood) exocytosis, as there are limited evidences about abluminal exocytosis and glycocalyx. Though, we have added the components of glycocalyx in the abluminal side (basement membrane) as suggested by the reviewer.
- The section on how GC modifies endocytosis could be better organized. For example, consider providing an overview statement describing the mechanisms by which GC components could impact endocytosis, transcytosis, and exocytosis. It might be worthwhile to add putative mechanistic information to one of the tables; for example, to indicate whether the impact of each GC component is by inducing membrane invaginations, engaging in low affinity charge-based or high-affinity ligand binding (e.g., syndecan:FGF), etc. The discussion of how GC modifies endocytosis (line 322) should follow directly after the description of the GC and would benefit from having a dedicated subheading. Discussion of Mucin-1 inducing membrane invaginations (line 150-152) could be moved to the section describing how GC modifies endocytosis/transcytosis (line 322-). What is the quantitative expression level of mucins in epithelial cells versus brain endothelial cells?
We appreciate the insightful suggestion of the reviewer. From our perspective, by introducing the BARs and glycocalyx in Section 2 and 3, we provide essential concepts needed to further understand the role of these elements in transcytosis. We then introduce transport across the brain endothelium and highlight the role of BAR and glycocalyx in the different steps of transcytosis. We believe that by organising the text in this way we provide general concepts, and then focus on the specific studies showing a role for glycocalyx in transcytosis.
Regarding the mucin-1, though the study from Shurrer et al. is relevant in terms of the role of glycocalyx in deformation of cell membranes, these studies demonstrated that mucins can generate forces to shape the membrane into finger-like extensions from the cell surface and not for mechanisms of endocytosis. For this reason, we mentioned mucins, but we kept them apart from the role of glycocalyx endocytosis. In terms of expression of mucins in brain endothelium versus epithelium, the reviewer raised a relevant point. The expression of mucins has been extensively studied in the epithelium, and it has been shown that these are expressed in the majority of tissues. However, very little is known about the expression of mucins in brain endothelium. Indeed, the expression of mucins in the brain has been mostly studied in the context of cancer, as the majority of cancers show high expression levels of mucins.
- Line 166: BEC do not undergo endocytosis; cargo undergoes endocytosis.
We have revised that detail in the legend of the figure 1.
- Line 159-160, 171-172: microglia are a subset of glia.
We have included microglia as a glial cell.
- Line 159-160, 171-172: oligodendrocytes are not commonly considered part of the BBB.
In line 167-170, we are describing the neurovascular unit that includes vascular cells (endothelial cells, pericytes and smooth muscle cells), glia (astrocytes, microglia, and oligodendrocytes) and neurons. We agree that microglia are a subset of glia and so we amended the text accordingly in line 169 and also in the legend of the Figure 1, line 156-157. When we describe the BBB, we do not include oligodendrocytes as part of the barrier and our reference to oligodendrocytes is as an element of the neurovascular unit.
- Line 197-198 unclear.
We have altered the sentence in line 196-198.
- Line 143 Table 2: In addition to citing the Human Protein Atlas, please also cite relevant primary literature describing the expression of proteoglycans in brain and non-brain endothelium.
We would like to thank the reviewer for the insightful suggestion. We have included relevant literature describing the expression of proteoglycans in non-brain endothelium, such as lung and umbilical vein endothelium. We have added the following sentences in line 132: “Apart from the brain endothelium, other endothelial cells express syndecans. GC in human lung microvasculature was confirmed by transmission electron microscopy, in which syndecans were described as essential proteins for the structure of the pulmonary endothelial barrier [37]. The human umbilical vein endothelial cells (HUVECs) have higher expression of syndecan-3 and -4 as compared with syndecan-1 and -2, contrary to brain endothelium [38].”
- Table 2. It is not clear why the authors are comparing brain and colon. Is the purpose of this comparison to show that the proteoglycans are broadly similar, with a few key differences (e.g. brain enrichment for Glypican 5-6), suggesting that Glypican 5-6 confers the suppression of transcytosis characteristic of the BBB relative to the colon? If yes, it would be helpful to specifically discuss how Glypican 5-6 influence transcytosis. If this does not reflect the authors’ intentions, greater clarification is warranted.
Colon endothelium was chosen as comparison because is the only non-brain endothelium described in Human Atlas Protein. We decided to include a sentence in the line 141 that compares brain and colon endothelium. We tried to access data from datasets published, particularly, recently published Single-Cell Transcriptome Atlas of Murine Endothelial Cells as well as other datasets (http://betsholtzlab.org/VascularSingleCells/database.html;https://markfsabbagh.shinyapps.io/vectrdb/). Taking in reviewer comment in consideration, we replaced the Table 2 with data based on Song et al. (1), in which RNA sequencing of the brain microvessels was performed in human and murine brains. We also added a sentence to refer to this Table in line 131.
- Line 205: why compare the human brain endothelium to the tick epithelium?
As mentioned in line 203-205, transcytosis is generally investigated in detail in the epithelium and/or peripheral endothelium, and it is accepted that similar mechanisms operate in the brain endothelium. Yet, brain endothelium is thinner than the epithelium and thus the cellular machinery (endosomes and lysosomes) may differ due to the reduced internal space. Here, we establish that comparison to further highlight the need to consider brain endothelium as distinct system and to reinforce the need to investigate transport specifically in BECs.
- Table 2 and 4: please align the text across columns.
We have aligned the text.
- Line 270 Table 4. Not clear why non-BBB cells (epithelium, fibroblasts) are included in this table, unless if adding discussion of BAR proteins in blood-CSF barrier or blood-meningeal barrier structures that contain epithelial cells and fibroblasts, respectively.
Along our manuscript we reinforced the lack of literature about the role of BAR proteins and glycocalyx in the brain endothelium. Thus, we attempted to bring all the evidence showing that these two elements are involved in mechanism(s) of transcytosis, even if in the epithelium, to establish the need for further studies investigating them in the brain.
- Please minimize excessive abbreviations which reduce readability. For example, the abbreviation CLIC/GEEC is introduced line 287 but not used again.
We have removed the abbreviation CLIC/GEEC in line 287.
- Line 296: Is there interplay between FEME and other CIE mechanisms, given the common deployment of some protein binding partners?
To the best of our knowledge, a clear interplay between FEME and other CIE mechanisms is yet to be described. Interestingly, Boucrot et al. (2)recently reviewed CIE mechanisms and raised similar questions. Core molecular machineries involved in CIE (such as, actin-polymerisation factors, BAR proteins and dynamin) may be used in a modular manner, depending on the cellular and/or physiological context. However, this modular manner needs to be further clarified to fill the gaps in knowledge that need to be addressed to complete our understanding of CIE.
- Figure 2. Unclear what the arrowheads and asterisk in a1 point indicate. The figure legend should be expanded to better explain the important (and beautiful) structural features in these images.
We added to the legend the following sentence: “Arrowheads point to an individual tubule for sFab and short elongated buds for dFab. Star shows vesicles with dFab signal within the lumen.”
- Line 232 typographical error: constrains, not constraints.
We corrected the typographical error. Please find alteration highlighted in yellow.
- Line 326. The phrase “endocytosis receptor” will be unfamiliar to a broad readership; please define.
We have altered the sentence to make it familiar to all readership.
- Line 328-331 contains two sentences that are unclear and appear to be contradictory.
We have rewritten the sentence to make it coherent: “In terms of proteoglycans, syndecans are involved in uptake of exosomes [47]. Besides, syndecan-4 acts as endocytic co-receptor by interacting with the fibroblast growth factor receptor 1 (FGFR1) for FGF2 uptake through a lipid raft-dependent, clathrin, and dynamin-independent manner through activation of small GTAse Rac1 [104].”
- Line 344 spelling error: emphasized.
We have altered the text.
References
- H. W. Song, K. L. Foreman, B. D. Gastfriend, J. S. Kuo, S. P. Palecek, E. V. Shusta, Transcriptomic comparison of human and mouse brain microvessels. Sci. Rep. 10, 12358 (2020).
- A. P. A. Ferreira, E. Boucrot, Mechanisms of Carrier Formation during Clathrin-Independent Endocytosis. Trends Cell Biol. 28, 188–200 (2018).
Please find in attachment the revised version of the manuscript.

Reviewer 2 Report
Overall, I think that the manuscript is well written and pretty comprehensive regarding the possible role of BAR proteins and glycocalyx in trafficking across the BBB endothelial cells. The review is well organized and easy to follow. The figures look nice.
main comments:
- In line 130/131, the human protein atlas is used to support the claim of expression of syndecan 2 and 4, but not 1 and 3 in brain endothelium. The authors should also consider using recent datasets where brain endothelial cells were transcriptomically profiled, which are available to search online (http://betsholtzlab.org/VascularSingleCells/database.html; https://markfsabbagh.shinyapps.io/vectrdb/)
- The manuscript usefully describes the features and functions of BAR-domain proteins and the existence of tubular vesicles in brain ECs. The review could be enhanced by greater discussion of how the properties of tubular (vs. spherical) transcytotic vesicles could contribute to BBB physiology. Is the higher surface area to volume ratio excluding bulk, fluid phase material that could otherwise be taken up non-specifically? They hint that it could help transcytotic cargo avoid lysosomal degradation, but it is not clear why that would be the case.
- Follow up with the previous point about unspecific uptake - the authors are focusing on the receptor-mediated transcytosis in the BECs and do not specifically discuss the possibility of non-selective transcytosis. For instance, - lines 207-208 – the authors state that “At BECs, transcytosis is mediated by the substrate-specific transport systems via receptor-mediated transcytosis”. This is ok and several ligand-receptor mediated transcytotic routes are indeed well established in the field. However, the authors should at least mention that there’s a possibility that non receptor-mediated transcytosis also occurs in BECs. Especially, considering that later in the manuscript (section 4.1.) they mention caveolae and CLIC/GEEC transcytotic routes, which are known to exhibit non-selective uptake.
- Lines 203-205 - The authors claim that in the cerebral endothelial cells the involvement of endosomes and lysosomes in the transcytosis across BBB “is less plausible considering that endothelium is very thin (c.a. 200 - 300 nm) compared to the tick epithelium (c.a. 10 μm)”. They cite Ayloo et al. review and another study from 90s by Keep et al. In my opinion based on these references, authors cannot claim that, but rather only suggest such a hypothesis. They either should reformulate the sentence or cite papers that actually show this observation (I am not aware of such studies).
- In section 4.1 Endocytosis – the authors present a comprehensive summary of clathrin-independent endocytic routes, including caveolae, CLIC/GEEC and FEME. I would suggest to include also the Arf6 mediated endocytosis and micropinocytosis, which are also known as clathrin-independent endocytotic pathways.
- Section 4.2. Intracellular Trafficking: Spherical versus Tubular Carriers – I would suggest elaborating a bit on the specific known molecular components of the endosomal compartments mentioned in this section (i.e. Rab5, Rab11, Lamp1, etc.) While sections 4.1 and 4.3 discussing endo- and exo-cytosis are very well explained, in my opinion this section requires more in-detail elaboration.
Minor comments:
- Line 205: “tick” should be thick?
- Reference 14 should be “De Camilli, P” not “Pietro, D.C.”
Author Response
Overall, I think that the manuscript is well written and pretty comprehensive regarding the possible role of BAR proteins and glycocalyx in trafficking across the BBB endothelial cells. The review is well organized and easy to follow. The figures look nice.
We would like to acknowledge the reviewer for the careful revision of our manuscript as well as for giving us the opportunity to revise it. We are pleased to address all comments. Please find below the response to each comment and alterations highlighted in yellow in a revised version of the manuscript.
In line 130/131, the human protein atlas is used to support the claim of expression of syndecan 2 and 4, but not 1 and 3 in brain endothelium. The authors should also consider using recent datasets where brain endothelial cells were transcriptomically profiled, which are available to search online (http://betsholtzlab.org/VascularSingleCells/database.html; https://markfsabbagh.shinyapps.io/vectrdb/)
We are grateful for the reviewer's suggestion. We agree that recent data should be mentioned, however most of the available datasets are from murine, including those recommended by the reviewer. In our perspective, since glycocalyx varies from specie to specie, specifically mice/humans, we prefer to highlight only the proteoglycans expression in humans but we added additional data on Table 2 showing expression in murine endothelial cells.
- The manuscript usefully describes the features and functions of BAR-domain proteins and the existence of tubular vesicles in brain ECs. The review could be enhanced by greater discussion of how the properties of tubular (vs. spherical) transcytotic vesicles could contribute to BBB physiology. Is the higher surface area to volume ratio excluding bulk, fluid phase material that could otherwise be taken up non-specifically? They hint that it could help transcytotic cargo avoid lysosomal degradation, but it is not clear why that would be the case.
While we highlight the relevance of tubular versus vesicular transport across BECs, there is still a limited amount of literature about the contribution of these structures to the BBB physiology. Initial studies by Bundgaard showed the intracellular trafficking through a network of tubules in BECs but there is little evidence in terms of receptor/ligands transported across those tubules. Recent evidence showed that in the case of two particular receptors (transferrin receptor (1) and LRP1 (2)), intracellular fate of the ligand/receptor depends on the avidity of the ligands (monovalent versusmultivalent). In both studies, it was also indicated that tubular-mediated transcytosis allows a faster transport of the cargoes across the BECs. In recent our study (2), we showed that, when LRP1 is transported across BECs via tubules, there is a decrease in the association of LRP1 with LAMP-1 (LAMP-1, a lysosomal marker). An opposite trend was found when LRP1 is transported via the vesicular endosomal pathway. Additionally, we demonstrated that by triggering the vesicular pathway, a reduction in the levels of LRP1 was observed within 1 hour, whereas the opposite was found when the tubular pathway was triggered by ligands for LRP1. Thus, based on this data, we propose that tubular pathway triggers a faster transport avoiding lysosomal degradation. Similarly, in the case of transferrin receptor, the tubulation-mediated transport shows to facilitate a faster transport avoiding degradation (1). Up to this date, there are no further studies that focused on tubular transport across the BECs. In our manuscript, we emphasised the relevance of these tubules in BECs to reinforce the need for further studies on this to understand the machinery involved and also receptors/ligands that may trigger such pathways.
- Follow up with the previous point about unspecific uptake - the authors are focusing on the receptor-mediated transcytosis in the BECs and do not specifically discuss the possibility of non-selective transcytosis. For instance, - lines 207-208 – the authors state that “At BECs, transcytosis is mediated by the substrate-specific transport systems via receptor-mediated transcytosis”. This is ok and several ligand-receptor mediated transcytotic routes are indeed well established in the field. However, the authors should at least mention that there’s a possibility that non receptor-mediated transcytosis also occurs in BECs. Especially, considering that later in the manuscript (section 4.1.) they mention caveolae and CLIC/GEEC transcytotic routes, which are known to exhibit non-selective uptake.
As mentioned above, the studies about tubular transport are focused on a receptor-mediated transport, in particular, transferrin receptor and LRP1. However, further work is needed to clarify the machinery and how these tubules are triggered by receptors/ligands. In our manuscript, we focused on substrate-specific transport at the BECs as this well established in the field and non-specific routes are not clearly studied in the brain endothelium. Along the manuscript, we bring all evidence about other endocytosis pathways in which BAR proteins or glycocalyx are involved to underline a possible role for these in the brain. So, even though we mention the CLIC/GEEC, we state that this pathway has not been identified in the brain endothelium and apart from the caveolar endocytosis with ageing, non-specific receptor-mediated transcytosis has not been established.
- Lines 203-205 - The authors claim that in the cerebral endothelial cells the involvement of endosomes and lysosomes in the transcytosis across BBB “is less plausible considering that endothelium is very thin (c.a. 200 - 300 nm) compared to the tick epithelium (c.a. 10 μm)”. They cite Ayloo et al. review and another study from 90s by Keep et al. In my opinion based on these references, authors cannot claim that, but rather only suggest such a hypothesis. They either should reformulate the sentence or cite papers that actually show this observation (I am not aware of such studies).
We have added further references. A particular relevant one is a study from Coomber et al. (3) in which it is stated a ~ 40% decrease in wall thickness in brain capillaries compared to non-cerebral capillaries.
- In section 4.1 Endocytosis – the authors present a comprehensive summary of clathrin-independent endocytic routes, including caveolae, CLIC/GEEC and FEME. I would suggest including also the Arf6 mediated endocytosis and micropinocytosis, which are also known as clathrin-independent endocytotic pathways.
In this review, we summarise the endocytic pathways that have been associated with BAR proteins and glycocalyx. We have added a few sentences about Arf6 mediated endocytosis as it has been described an interaction between endophilin and Arf6. Please find the alterations highlighted in a revised version of the manuscript (line 288-293).
- Section 4.2. Intracellular Trafficking: Spherical versus Tubular Carriers – I would suggest elaborating a bit on the specific known molecular components of the endosomal compartments mentioned in this section (i.e. Rab5, Rab11, Lamp1, etc.) While sections 4.1 and 4.3 discussing endo- and exocytosis are very well explained, in my opinion this section requires more in-detail elaboration.
We have added a few details about the endosomal trafficking in section 4.2.
- Line 205: “tick” should be thick?
We have corrected the typographical error.
- Reference 14 should be “De Camilli, P” not “Pietro, D.C.”
We corrected the reference accordingly.
References
- R. Villasenor, M. Schilling, J. Sundaresan, Y. Lutz, L. Collin, Sorting Tubules Regulate Blood-Brain Barrier Transcytosis. Cell Rep.21, 3256–3270 (2017).
- X. Tian, D. Leite, E. Scarpa, S. Nyberg, G. Fullstone, J. Forth, D. Matias, A. Apriceno, A. Poma, A. Castano, M. Vuyyuru, L. Harker-Kirschneck, A. Saric, Z. Zhang, P. Xiang, B. Fang, Y. Tian, L. Luo, L. Rizzello, G. Battaglia, On the shuttling across the blood-brain barrier via tubules formation: mechanism and cargo avidity bias. bioRxiv (2020), doi:10.1101/2020.04.04.025866.
- B. L. Coomber, P. A. Stewart, Morphometric analysis of CNS microvascular endothelium. Microvasc. Res. 30, 99–115 (1985).
Please find the revised version of the manuscript in attachment.
